# Vascularization of Poly-ε-Caprolactone-Collagen I-Nanofibers with or without Sacrificial Fibers in the Neurotized Arteriovenous Loop Model

**DOI:** 10.3390/cells11233774

**Published:** 2022-11-25

**Authors:** Simon Kratzer, Andreas Arkudas, Marcus Himmler, Dirk W. Schubert, Dominik Schneidereit, Julian Bauer, Oliver Friedrich, Raymund E. Horch, Aijia Cai

**Affiliations:** 1Department of Plastic and Hand Surgery and Laboratory for Tissue Engineering and Regenerative Medicine, University Hospital of Erlangen, Friedrich-Alexander University Erlangen-Nürnberg (FAU), 91054 Erlangen, Germany; 2Institute of Polymer Materials, Department of Materials Science and Engineering, Friedrich-Alexander University Erlangen-Nürnberg (FAU), 91058 Erlangen, Germany; 3KeyLab Advanced Fiber Technology, Bavarian Polymer Institute, Dr.-Mack-Strasse 77, 90762 Fürth, Germany; 4Institute of Medical Biotechnology, Department of Chemical and Biological Engineering, Friedrich-Alexander University Erlangen-Nürnberg (FAU), 91052 Erlangen, Germany

**Keywords:** vascularization of nanofiber scaffolds, neurotization, neoangiogenesis, EPI loop model, AV loop model, PCL-collagen I-nanofiber scaffolds, polyethylene oxide

## Abstract

Electrospun nanofibers represent an ideal matrix for the purpose of skeletal muscle tissue engineering due to their highly aligned structure in the nanoscale, mimicking the extracellular matrix of skeletal muscle. However, they often consist of high-density packed fibers, which might impair vascularization. The integration of polyethylene oxide (PEO) sacrificial fibers, which dissolve in water, enables the creation of less dense structures. This study examines potential benefits of poly-ε-caprolactone-collagen I-PEO-nanoscaffolds (PCP) in terms of neovascularization and distribution of newly formed vessels compared to poly-ε-caprolactone -collagen I-nanoscaffolds (PC) in a modified arteriovenous loop model in the rat. For this purpose, the superficial inferior epigastric artery and vein as well as a motor nerve branch were integrated into a multilayer three-dimensional nanofiber scaffold construct, which was enclosed by an isolation chamber. Numbers and spatial distribution of sprouting vessels as well as macrophages were analyzed via immunohistochemistry after two and four weeks of implantation. After four weeks, aligned PC showed a higher number of newly formed vessels, regardless of the compartments formed in PCP by the removal of sacrificial fibers. Both groups showed cell influx and no difference in macrophage invasion. In this study, a model of combined axial vascularization and neurotization of a PCL-collagen I-nanofiber construct could be established for the first time. These results provide a foundation for the in vivo implantation of cells, taking a major step towards the generation of functional skeletal muscle tissue.

## 1. Introduction

The engineering of three-dimensional (3D) skeletal muscle tissue constructs holds promise for treating volumetric muscle loss without sacrificing healthy donor muscle [1,2,3]. In relation to this, the in vivo production of tissue represents a promising option [4,5,6,7]. Axial vascularization enables a sufficient vascular supply and is therefore essential for the survival of transplanted and engineered tissue [8,9]. As early as 1980, the microsurgical connection of an artery and a vein by means of an interposed vein interposition, and a resulting arteriovenous (AV) loop was described in this context [10]. Implanting this AV loop into a matrix enclosed by an isolation chamber enables axial vascularization of the implanted matrix by sprouting new vessels from the loop, a process known as intrinsic vascularization. The resulting neo-tissue can be transplanted using the vascular axis of the AV loop, which is directly connected to local vessels at the recipient site. The resulting systemic blood circulation, allowing gas and nutrient exchange as well as the transport of metabolic products, enables direct integration of the implant into the recipient organism [8].

In addition to vascularization, the carrier matrix also plays an important role in tissue engineering [11]. Since skeletal muscle represents a complex tissue with hierarchically organized fibers, a matrix, mimicking those properties is needed [12]. Electrospinning is a robust technique to produce aligned nanofibers which mimic the structural anisotropy of myofibres and their extracellular matrix [12,13,14].

Many biodegradable polymers such as poly-ε-caprolacton (PCL) have shown both in vitro and in vivo biocompatibility [1]. Co-spun with collagen, this polymer has induced myogenic differentiation of primary myoblasts and mesenchymal stem cells, even after long-term cultivation [1]. However, nanoscaffolds often consist of high density packed fibers, resulting in poor cell infiltration. This results in regions of acellularity within the scaffold interior [15]. Thus, it is of great importance to create cell-permeable electrospun scaffolds with enhanced cell distribution and migration throughout the scaffolds’ dimension [13]. The water-soluble polymer polyetyhlene oxide (PEO) has been shown to increase porosity of PCL-collagen I-nanoscaffolds (PC) as sacrificial fibers, leading to improved distribution of mesenchymal stem cells within the scaffold [13].

Three-dimensional vascularization patterns by means of the AV loop differ in nanofiber scaffolds with different architectures [16]. Aligned PCL-collagen I-PEO-nanoscaffolds (PCP) have presented a smaller number of sprouting vessels while vascularization occurred considerably earlier in the center of the scaffolds compared to randomly spun PC [16]. However, as stated by the authors, a direct comparison between aligned and randomly spun nanofibrous scaffolds was difficult [16].

To generate functional muscle tissue, nerval stimulation by, for instance, integration of a motor nerve is crucial. A further development of the AV loop model, the EPI loop model links axial vascularization with the integration of a motor nerve branch enabling neurotization of an axially vascularized scaffold [17]. This nerve serves as a myogenic stimulator, promoting myogenic differentiation of implanted muscle progenitor cells [17].

In this study, we aimed to compare PC to PCP in terms of vascularization, neurotization and cell interaction after implantation into the EPI loop model. To enable direct comparison between both groups, aligned nanofibers as representatives of the physiological arrangement of myofibres were used for both scaffold types in contrast to previous research.

## 2. Materials and Methods

### 2.1. Fabrication and Characterization of Nanoscaffolds

PC and PCP were produced by electrospinning as previously described [1,18]. Briefly, PCL (80.000 g/mol, Sigma Aldrich, St. Louis, MO, USA) was blended with bovine collagen type I (Symatese, Lyon, France) in a ratio of 2:1 at a 12% (*w*/*v*) solution, using 90% acetic acid (Carl Roth GmbH, Karlsruhe, Germany) as a solvent. PCL-collagen I-nanofibers were electrospun on a standard electrospinning machine onto parallel metal rods on a custom-made rotating drum collector (15 kV, 15 cm, 1 mL/h, 50 rpm). PEO (concentration 10% (*w*/*v*), molecular weight: 900.000 g/mol, Sigma Aldrich) nanofibers were similarly spun (14 kV, 13 cm, 1 mL/h, 50 rpm). The aligned fibers were collected in alternating layers on polyamide rings with an inner diameter of 8 mm and 0.3 mm of height (A.R.T. Elektromechanik GmbH, Munich, Germany).

Both PC and PCP were soaked in 70% ethanol (EtOH) over night and were afterwards rinsed with phosphate-buffered saline (PBS), three times for five minutes each before further experiments. The morphology of the scaffolds before and after pretreatment with EtOH and PBS was analyzed via scanning electron microscopy (SEM, Auriga Fib, Zeiss, Oberkochen, Germany). For this purpose, they were sputter-coated with gold using a Q150T Turbo-pumped Sputter Coater (Quorum Technologies Inc., Guelph, ON, Canada) [18]. Fiber diameter was measured using ImageJ (National Institutes of Health, Bethesda, MD, USA, Version 1.53e) and fiber orientation was defined using the ImageJ plugin OrientationJ.

To detect potential PEO residues after scaffold pretreatment, thermo-gravimetric analysis (TGA) was performed on a TGA Q5000 (TA Instruments). The scaffolds as well as the single components were heated from room temperature to 600 °C at a heating rate of 10 K/min under a constant nitrogen flow. However, as decomposition temperatures of the blend materials are similar, detection of PEO residues by TGA turned out to be inappropriate (Appendix A). Thus, Fourier-transform infrared spectroscopy (FT-IR) analysis was carried out. For this, the 70% ethanol and the three PBS washing solutions as well as the pure liquids were collected and dried over night at 80 °C. Then, the individual residues were mixed with potassium bromide (KBr) powder to produce discs for FT-IR analysis. For comparison, the same procedure was followed with PEO. FT-IR measurements were conducted from 4000 cm^−1^ to 400 cm^−1^ using a Nicolet 6700 FTIR spectrometer (Thermo scientific, Waltham, MA, USA).

Elastic properties of the aligned PCL-collagen I- and PCL-collagen I-PEO-nanofibers were determined using a single fiber tensile testing machine (Vibrodyn 400, Lenzing instruments GmbH & Co. KG, Gampern, Austria) as described before [19]. Briefly, the aligned fibers were formed into a fiber bundle, and the mechanical properties of the respective bundle were measured. Post-measurement, the weight of the sample is registered and with the knowledge of materials density, the elastic properties of a single fiber can be calculated.

### 2.2. Experimental Groups

PC or PCP were implanted into five rats, respectively over a period of two (PC2 resp. PCP2) or four (PC4 resp. PCP4) weeks for each group. All groups underwent the same surgical and explantation procedure, and thus differ only in the duration of the experiment and the nature of the implanted nanofiber scaffolds.

After each implantation period, loop vessels were perfused with India ink (Lefranc-Bourgeois, London, UK), and constructs were explanted for later immunohistochemical analysis (*n* = 4). Multiphoton microscopy was performed on one construct per group and time period for visualization of the loop and its neovessels (*n* = 1). To enable statistical analysis, rats showing signs of vessel thrombosis were excluded from the study and replaced by newly operated animals. Thus, a total of 27 animals were operated for this study.

### 2.3. EPI Loop and Scaffold Implantation

Animal experiments were approved by the animal care committee of the Friedrich-Alexander University of Erlangen-Nürnberg and the Government of Mittelfranken, Germany (approval numbers: RUF-55.2.2-2532-2-161-71 and RUF-55.2.2-2532-2-1315-15) and were carried out according to the German regulations for the care of laboratory animals at all times. Male T cell-deficient Rowett Nude (RNU) rats, weighing 310-450 g underwent surgery under anesthesia, using a vaporizer for anesthetic gases (Penlon, Sigma Delta Vaporizer, Penlon Ltd., Abingdon, UK) including isoflurane (Isoflurane CP^®^, 1 mL/mL, CP Pharma, Burgdorf, Germany). The surgery was performed under a surgical microscope (Carl Zeiss, Jena, Germany) by the same surgeon (S.K.). Prior to surgery, the surgical area was shaved and disinfected.

In a preliminary study, it became apparent that the previously described EPI loop model, using the saphenous artery and the superficial inferior epigastric vein (SIEV) as loop vessels [20] results in an unfavourable entry angle of the artery when the chamber is rotated medially. However, since the process of rotation is mandatory when implanting the used motor branch of the obturator nerve, an alternative arterial inflow had to be developed. By using the superficial inferior epigastric artery (SIEA) instead of the saphenous artery as the arterial limb of the EPI loop, rotation of the chamber opening can be enabled to integrate the motor nerve branch into the model, while ensuring an improved angle of the entering vascular pedicle into the implantation chamber (Figure 1).

For the surgical procedure, a semi-curve incision was made in the left groin while a straight transverse incision was made in the right groin to expose the right SIEV as a vein graft. The left superficial inferior epigastric vessels were localized and freed from the surrounding connective and fatty tissue. The proximal end of the right SIEV was anastomosed to the dissected left SIEV, using interrupted 11-0 microsutures (11-0 Ethilon™, Ethicon, Inc., Raritan, NJ, USA). Subsequently, the left SIEA was also dissected and anastomosed to the distal end of the venous interposition graft as described above, creating the arterio-venous anastomosis. Before opening the clamp attached to the arterial leg, 25 I.U. heparin (Heparin sodium 25000-ratiopharm^®^, ratiopharm GmbH, Ulm, Germany) were applied intravenously via the tail vein. After removal of the clamp, loop patency was checked, using the ballooning test, excluding retrograde perfusion.

Additionally, the left obturator nerve including its nerve branches was dissected from the adductor muscles as described previously [17]. A sterilized polytetrafluoroethylene (PTFE) isolation chamber with 1 cm in diameter and 0.6 cm in height, and an opening for the entrance of the loop and nerve, was filled with two layers of nanofiber scaffolds of either PC or PCP. Four spacers (stylet for Vasofix^®^, G 18 × 45 mm, green; B. Braun Melsungen AG, Melsungen, Germany) were implanted in the center for subsequent loop and nerve implantation and fixation (Figure 2A). The whole construct was placed into the left groin and fixed to the underlying muscle. The EPI loop was placed around the four spacers on top of the scaffolds while the most proximal obturator nerve branch was fixed to the spacer, being most distant from the opening of the chamber so that the nerve lied in the middle of the EPI loop (Figure 2B). Care had to be taken to pass the nerve suture only through the epineurium to avoid damaging the axonal structures of the nerve. The EPI loop and the nerve were then embedded in fibrin gel (20 mg/mL fibrinogen in saline solution mixed with thrombin 4 IU/mL in 40 mM calcium-chloride solution, Tisseel VH/SD, Baxter Healthcare S.A., Wallisellen, Switzerland) as described previously [16] (Figure 2C). The chamber was filled with two more layers of nanoscaffolds as described above, filled with another layer of fibrin (Figure 2D) and finally the lid was closed with a cap. The construct was fixed to the underlying muscle and the skin was closed [17]. All operated animals were treated postoperatively for five days with enrofloxacin (Baytril^®^, 25 mg/mL, Bayer Vital GmbH, Germany), enoxaparin sodium (Clexane^®^ multidose, 100.000 I.E./10 mL, Sanofi), and meloxicam (Metacam^®^ 2 mg/mL, Boehringer Ingelheim Vetmedica GmbH, Germany). Overall, the animals tolerated the procedures well.

### 2.4. Loop Perfusion and Explantation of Constructs

Four constructs per group per time period were used for immunohistochemical analysis. Vascularization was visualized after intra-aortal perfusion with India ink (Indian Black Ink, LeFranc & Bourgeois, London, UK) as described previously [21]. Briefly, rats underwent a longitudinal laparotomy. The descending aorta was cannulated and the inferior caval vein was perforated. 150–200 mL of a heated Ringer-Heparin solution (100 IU/mL) were used to flush the aorta until the blood was washed out of the system. 30 mL of India ink (50% *v*/*v*) in 5% gelatin (Carl Roth, Karlsruhe, Germany) and 4% mannitol (Carl Roth) was injected into the arterial system. After ligation of the aorta and the inferior caval vein, the specimens were placed at −20 °C for 1–2 h. Afterwards, the constructs including the scaffolds and the EPI loop were taken out of the isolation chambers, weighed and then placed in 4% formalin (Carl Roth) over night for paraffin embedding.

### 2.5. Immunohistochemical Analysis

After formalin-fixation, the constructs were cut perpendicular to the axis of the AV loop. The halves of the constructs were paraffin embedded. The half including the vessels entering the loop was referred to as “proximal” while the other half, containing the vein graft was referred to as “distal”. 3 µm cross-sections were cut of each half and stained with hematoxylin and eosin (HE) and Masson’s Trichome according to standard protocols. Vessels were visualized with α-smooth muscle actin (SMA) staining (primary antibody: mouse monoclonal antibody actin smooth muscle, Zytomed Systems, Berlin, Germany; secondary antibody: conjugation of anti-mouse (Ig) and anti-Rabbit (Ig), Life Technologies, Carlsbad, CA, USA) [22]. Methylenblau, Synaptophysin immunofluorescence (anti-Synaptophysin antibody, Abcam, Cambridge, UK) and S100 immunohistochemial (S100 monoclonal antibody, Life Technologies) stainings were used for visualization of the implanted nerve. For macrophage detection, including pro- (M1-) and anti-inflammatory (M2-) macrophages CD86 (Abcam) and CD163 (Leica Biosystems Inc., Deer Park, IL, USA), immunostainings were performed as previously described [21]. After completing the staining, all sections were photographed under 10× magnification, using an Olympus IX81 (Olympus, Hamburg, Germany), and then stitched together, using the software cellSens Dimension V.1.5.

### 2.6. Quantification and Statistics

Quantification was performed in a blinded fashion. In order to obtain comparable results, the histological sections of the individual preparations were each performed at corresponding, previously defined locations within the construct.

Newly formed vessels were counted in α-SMA stained sections. Using ImageJ 1.53e (National Institutes of Health, Bethesda, MD, USA), α-SMA positive vessels with a lumen or lumens filled with India ink were counted and a coordinate was assigned to each of them [21,22]. On the basis of those parameters, the respective distance to the corresponding lumina of the main loop was determined. M1 and M2 macrophages were counted in anti-CD86 and anti-CD163 stained sections, respectively. Using ImageJ 1.53e the total number of pro- and anti-inflammatory macrophages was quantified [21]. Statistical analysis was performed with GraphPad Prism 8.0 (GraphPad Software, San Diego, CA, USA). With regard to the number and the distances of newly formed vessels, the mean values of all sections were statistically evaluated. Normal distribution of the data was confirmed with Shapiro–Wilk-test. An unpaired *t*-test, paired *t*-test, Mann–Whitney test or Wilcoxon matched-pairs signed rank test was performed as appropriate. A *p* value ≤ 0.05 was considered statistically significant.

The weights of the explanted constructs were measured and hereby, the degradation of the fibrin matrix was evaluated.

### 2.7. Multiphoton Microscopy

A native construct of PC implanted into the EPI loop with subsequent explantation was used for establishing the visualization of the EPI loop and the nerve via multiphoton microscopy. Afterwards, one formalin fixed construct per group and per time period was explanted and dehydrated and optically cleared as described previously [21,23]. The overview 3D mosaics were acquired with a voxel size of 4.3 × 4.3 × 4 µm^3^, each mosaic piece having a field of view of 1.1 × 1.1 × 2.2 mm^3^. The stitching of the mosaic was performed using the software Fiji [24] and its stitching plugin [25]. High-resolution 3D stacks of confined regions of interest were acquired with a voxel size of down to 1 × 1 × 1 µm³ and fields of view adapted to the feature size. An upright Trimscope II with a setup similar to the one described by Schneidereit et al. [26] was used for multiphoton microscopy, using a femtosecond pulse titanium-sapphire LASER, tuned to a wavelength of 810 nm as excitation light source. The backscattered Second Harmonic Generation (SHG) signal was detected using a bandpass filter with a 405 nm centre wavelength and 20 nm bandwidth. Two autofluorescence bands were recorded, one at 525 nm with 25 nm width and one at 620 nm with 60 nm width. 3D reconstructions to visualize the volumetric image were performed using the Fiji 3D viewer [27].

## 3. Results

### 3.1. Characterization of Nanofibers

Microstructural analysis of the nanofiber scaffolds revealed that compared to PC, intermediate gaps were formed in PCP as a result of dissolving out the PEO (Figure 3). Before treatment, PC had a mean diameter of 554 nm ± 332 nm while PCP had a mean diameter of 233 ± 116 nm. The determination of the modulus of elasticity resulted in 20.9 ± 7.2 MPa for PC and 17.2 ± 5.0 MPa for PCP.

The fiber orientation is given as standard deviation from the mean direction, meaning that 68.27% of all fibers are aligned within this range. For PCP, a standard deviation from the mean direction of 4.6° was determined. For a 2σ standard deviation, including 95.45% of all fibers, a deviation from the mean direction of 9.2° could be observed. A slightly lower standard deviation from the mean direction could be measured for PC with σ = 2.3° and 2σ = 4.6°.

After the pretreatment with EtOH and proceeding washing steps with PBS, a reduced orientation of the nanofibers could be observed (Figure 3). Scaffolds, containing PEO nanofibers showed a standard deviation from the mean direction of 8.1°. For PC, a standard deviation from the mean direction of 4.7° was measured. Fibers agglomerated in the washing process and subsequently fiber bundles were formed. With the removal of the sacrificial PEO nanofibers, the free volume allowed for more movement of the nanofibers, counteracting the fiber alignment. However, SEM analysis exposed a less dense structure for PCP (Figure 3).

FT-IR analysis of the PBS solutions revealed no characteristic peak of PEO in all three washing rounds. All PBS solutions used for washing the scaffolds showed identical patterns as the clear PBS solution, therefore, showing no residues of PEO. In contrast, explicit peaks of PEO could be detected in the EtOH solution. From these results, it can be concluded that PEO is completely washed out during the sterilization process (Appendix A).

### 3.2. Patency Rate of the EPI Loop

Patency of the loop vessels was assessed macroscopically immediately before perfusion and histologically based on the filling of the lumina with India ink as well as the presence of newly formed vessels as a sign of neovascularization. Of the 27 operated animals, 16 animals showed a patent loop (patency rate of 59.26%) based on macroscopical and histological evaluation (Table 1). For histological evaluation, newly formed vessels around the loop vessels could be identified as a sign of neovascularization [28]. Four animals per group (n = 4) were evaluated in the two- and four-weeks groups. PC (patency = 60.71%) showed a lower thrombosis rate than PCP (patency = 53.33%).

### 3.3. Explant Weights

There was no difference between the explant weights of PC2 (0.4725 g ± 0.1089 g) and PCP2 (0.5425 g ± 0.1249 g) (*p* = 0.25) (t(6) = 0.7313, *p* ≤ 0.05). Comparing PC4 (0.3325 g ± 0.0083 g) and PCP4 (0.42 g ± 0.0453 g), a significantly higher loss of weight (*p* = 0.0083) was detected in the PC group (t(6) = 3.293, *p* ≤ 0.05). In the PC group, a significant decrease in the explant weights (*p* = 0.0476) was observed with longer implantation period (two vs. four weeks) (t(3) = 2.407, *p* ≤ 0.05). Similarly, PCP showed a statistically significant (*p* = 0.046) decrease in the explant weights over time (t(3) = 2.447, *p* ≤ 0.05) (Figure 4).

### 3.4. Vessel Quantification and Distribution

The newly formed vessels could be identified by the incorporation of α-SMA. The injected India ink accumulated only in the vessel lumen but not in the extravascular space, suggesting adequate stability and maturity of the newly formed vessels (Figure 5).

After two weeks of implantation, neovascularization could be identified for both scaffold types. Comparing PC2 (228.25 ± 351.21 vessels) and PCP2 (115.75 ± 108.15 vessels), no significant difference (*p* = 0.28) could be detected with respect to the number of new vessels (t(6) = 0.6123, *p* ≤ 0.05). After four weeks, the PC4 (527.25 ± 313.17 vessels) showed higher neoangiogenesis compared to PCP4 (138.5 ± 85.91 vessels) (*p* = 0.0443) (t(6) = 2.030, *p* ≤ 0.05). A significant increase in neo-angiogenesis could be observed from two to four weeks of implantation for both scaffold types. In the PC group, an increase in the number of vessels was observed with longer implantation period (*p* = 0.0312) (t(3) = 2.903, *p* ≤ 0.05). Similarly, PCP showed a significant increase in newly formed vessels when comparing PCP4 with PCP2 (*p* = 0.0081) (t(3) = 4.909, *p* ≤ 0.05) (Figure 6A).

No significant differences were found in the vessel distances between the different types of scaffolds after two weeks (*p* = 0.5) and four weeks (*p* = 0.1752). Regarding the implantation duration, no significant increase in the vessel distances to the main lumen could be observed within PC4 (0.758mm ± 0.272mm) compared to PC2 (0.463mm ± 0.391 mm) (*p* = 0.0684) (t(3) = 2.019, *p* ≤ 0.05). Similarly, within the PCP group, no significantly larger distance was found between PCP4 (0.54mm ± 0.254mm) and PCP2 (0.62mm ± 0.254mm) (*p* = 0.4375) (Figure 6B).

### 3.5. Assessment of Immunological Activity

Invasion of both pro-inflammatory (M1) and anti-inflammatory (M2) macrophages into the construct was detected in the corresponding stains (CD86 and CD163) (Figure 7). After two weeks, no difference was observed between PC2 (9691 ± 9729) and PCP2 (5718 ± 5147) with regard to invasion of M1 macrophages (*p* = 0.2775) (t(6) = 0.6251, *p* ≤ 0.05). Similarly, no difference in M1 count between PC4 (4179 ± 2798) and PCP4 (9866 ± 6745) could be detected after four weeks (*p* = 0.1131) (t(6) = 1.349, *p* ≤ 0.05). In both PC (*p* = 0.1316) (t(3) = 1.374) and PCP (*p* = 0.0886) (t(3) = 1.757), no statistical difference was detected between two and four weeks of implantation. The number of M2-macrophages did not differ between PC2 (364 ± 354) and PCP2 (103 ± 154) (*p* = 0.1714) as well as between PC4 (376 ± 206) and PCP4 (359 ± 374) (*p* = 0.4742) (t(6) = 0.0674, *p* ≤ 0.05). In both PC (*p* = 0.4582) (t(3) = 0.1124) and PCP (*p* = 0.0625), no statistical difference could be detected between two and four weeks of implantation.

In all four groups, the invasion of cells between the nanofibers was observed. Depending on the degree of vascularization, migration was more or less accentuated (Figure 8).

### 3.6. Multiphoton Microscopy

Multiphoton microscopic evaluation of the EPI loop PC construct explanted immediately after surgery revealed a strong signal from both the vascular loop and the motor nerve branch in the Second Harmonic Generation channel (Figure 9). The images in multiple slices allow a three-dimensional processing of all layers, illustrating the EPI loop and nerve in relation to the scaffolds and matrix (Appendix A).

Within the four constructs analyzed, neovascularization could be identified only in the PC4 construct. Newly formed vessels in the vicinity of the main vessels (Figure 9E) could be detected and visualized. Neovascularized areas showed tight capillary structures (Figure 9C).

### 3.7. Assessment of Nerval Structures

To demonstrate the viability of the nerve branch in the constructs, different nerve-specific structures were examined. The integration and vascularization of the implanted nerve branch into the loop model could be observed histologically. Using special staining with methylene blue, ganglia cells could be stained (Figure 10A). Using S100 immunohistochemistry, the presence of peripheral glia cells resp. Schwann cells was detected, confirming the stability of the neural structures within the EPI loop model (Figure 10D).

## 4. Discussion

With the results of this study, a novel EPI loop model could be established for the first time, which integrates the SIEA into the model instead of the saphenous artery which has been previously used for the EPI loop model [17]. While previous studies separately compared the difference between PC and PCP scaffolds or the influence of axial vascularization connected to neurotization, now these findings were investigated in combination, also for the first time. Furthermore, aligned nanofibers were used in both groups for the purpose of skeletal muscle fabrication as well as for appropriate comparison of the two different scaffold types. In both groups, incipient vascularization of the constructs could already be observed after two weeks.

The comparison of the two groups after four weeks of implantation must be considered in a differentiated manner. Despite the PEO fibers being washed out and the resulting free spaces, the statistical evaluation showed a stronger neovascularization in the PC4 group. The pronounced weight loss in PC4 compared to the PCP4 also indicates a higher amount of vascularization, cell migration, and maturation in the matrix as suggested by Schmidt et al. [29]. These results are contrary to the assumption that loosening of the fibrous structure of the scaffolds in PCP would favor ingrowth of vessels. Similar observations were described by Klumpp et al. [16]. However, randomly arranged PCL-collagen I-nanofibers were compared with parallel arranged PCP in that study while the current study was the first to compare aligned nanofibers in both PC and PCP, leading to similar results as previously described. The hypothesis put forward by Klumpp et al. that those differences were due to the different alignment of the nanofibers [16] therefore seems rather unlikely, and other factors should be considered to explain the stronger vascularization of the more densely arranged scaffolds. Two physiological conditions provide a possible explanation for the results contrary to our hypothesis. First, smaller pore diameters per se do not represent an obstacle to cell migration [30]. Second, the required mechanical stability of the matrix could be lost due to the enlarged pores. Previously described by Zhang et al., there is the possibility for cells to grow into nanofiber constructs whose fiber diameters are larger than the distances between the fibers. The biomimetic nanofibers provide a microenvironment, analogous to a natural extracellular matrix. Moreover, they deliver adjusted mechanical response for cell movement, allowing cells to penetrate the scaffolds through amoeboid movements and migrate through the gaps, pushing the contiguous fibers aside to expand the diameter of the gaps, so that the ingrowth of vessels into denser structures can also be partially explained [30]. Taking into account the results of our study, as well as the multitude of tissue-cell interactions that include mechanical stimuli and thereby influence the morphogenesis of cell assemblies [8], one must also consider that the removal of sacrificial fibers might, on the one hand, create free space for the hypothetical ingrowth of vessels but, on the other hand, affect fiber morphology and mechanical properties of the scaffolds and therefore affect the interactions between the growing vessels and the surrounding tissue. Although porosity is crucial for cell migration in scaffolds, increasing distance between fibers can adversely affect tissue growth. Increasing pore size is an obstacle when cells try to bridge the gap or form functional tissue [31]. A recent study on the kinetics of cell bridging on scaffolds showed that with increasing pore diameter, cell colonization occurred both later and less pronounced [32]. In our case, the detachment of the sacrificial fibers, which aimed to create more space, may have exceeded a critical point at which these necessary mechanical properties of the scaffolds were partially lost. Rather, the densely packed PC could provide a more mechanically stable system for the differentiating cell assemblies and provide cell-to-cell-contact [33], thereby driving increased and faster neoangiogenesis.

In addition, an interesting finding from the exploration of other scaffold types is a possible correlation between the stiffness of the implanted materials and their neovascularization [29,34]. Therefore, we evaluated the modulus of elasticity of untreated PC and PCP, showing only a small difference of 4 MPa. However, in this study, since the PEO fibers were completely washed out as shown by FT-IR, there should be only PC fibers left after treatment. Nonetheless, a reduction of the stiffness of the PCP scaffold is to be expected as the same force is applied to a less dense area compared to PC scaffolds. Since higher stiffness correlates with neoangiogenesis [29], this phenomenon could also explain the results of our study.

As a possible solution to the problems mentioned here, in future studies in which the scaffolds are seeded with cells, modified nanofibers that are modulated in their mechanical properties could combine improved mechanical interaction with larger and more permeable pores, thereby further improving cell migration. Anyway, the results of this study prove that more porous scaffolds are not inevitably superior to denser fibers.

Overall, it must be mentioned as a limitation that no three-dimensional imaging was performed in this study, which could be used to assess vascularization more adequately than is the case using two-dimensional paraffin sections. Two-dimensional sections only image one level of the construct at a time, and are limited in their ability to image the parameter of differential occurrence of neovascularization [16]. To limit this bias, both proximal and distal sections were evaluated and the mean values of the data obtained were calculated. However, in order to obtain an optimal result, a high number of histological sections representing each section of the specimen would be necessary. Considering the time and technical effort of its implementation, the question arises whether there are not generally more efficient procedures for this, such as the implementation of micro-CT or multiphoton microscopy (for which however, quantitative analyses are not yet commonly established). In general, not only the assessment of the absolute number of new vessels, but also a better qualitative evaluation of the spatial distribution of structures would be possible, which is certainly extremely relevant in the context of the implantation of three-dimensional cell constructs. Furthermore, it must be mentioned that exact evaluation of corresponding sections of different constructs was made difficult due to inaccuracies during sectioning. This led to a rather high standard deviation of the counted vessels and cells.

With the help of multiphoton microscopy, the newly developed EPI loop model could be visualized three-dimensionally, and the occurrence of neovascularization could also be detected. However, because only one construct per group was used, only limited conclusions can be drawn. Nevertheless, the results seem to be in agreement with those of the immunohistochemistry, showing a stronger neovascularization in PC4. Here, the question arises whether the method of multiphoton microscopy is not as sensitive as that of immunohistochemistry, since a certain amount of vessels must first be formed before they can be visualized in the imaging. An interesting option for further studies would be a quantitative evaluation of the vessels as well as a qualitative visualization of the structures. In addition to the expected more accurate results, this would also subtract the occurrence of artifacts and distortions of the structures, which inevitably arise in the histological evaluation due to the invasive preparation of the sections.

The observed patency rate of the newly developed EPI loop of about 60% is in line with the results of previous studies using the AV loop [8,16,17]. It has to be taken into account that the EPI loop vessels have a smaller lumen than the AV loop vessels, and therefore the EPI loop model is generally more prone to thrombosis than the classical AV loop model. Furthermore, compared to studies in which only one AV loop was implanted into a matrix [16], it must be noted that the implantation of four layers of nanofiber scaffolds with an additional motor nerve naturally represents an additional obstacle, and could potentially affect the thrombosis rate of the main vessels. Thus, a comparison of the different loop models concerning patency rate is only possible to a limited extent.

In terms of nanofiber biocompatibility, both scaffold types were well tolerated and showed no difference in the immunological activity as indicated by macrophage invasion, endothelial cell interactions, and integration into the situs.

The assessment of the explant weights and the degradation of the fibrin matrix that can be derived from this showed a greater degradation in PC than PCP after four weeks of implantation. These results can be related to the results of vessel quantification. According to this, increased vascularization allows enhanced degradation of the matrix, due to boosted migration of cells. With regard to the choice of matrix, these results, which already show a significant reduction of the matrix after four weeks, raise the question of whether sufficient stability of the model can be maintained for longer implantation periods in which cells are inserted into the system. Nevertheless, the compatibility of fibrin with respect to myogenic differentiation of MSC after an implantation period of eight weeks has already been demonstrated [17,28], which certainly justifies the preference of this matrix over other more stable matrices.

## 5. Conclusions

With the establishment of the newly developed EPI loop model, the elements of a sufficient nutrient supply through the superficial inferior epigastric vessels, a physiological extracellular structure through parallel aligned nanofiber scaffolds, and the central element of neurotization through a branch of the obturator nerve can be linked. Here, comparison of the differences in vascularization of PCL-collagen I and PCL-collagen I-PEO provides important data for a clinical application of the model. The use of PCL-collagen I-scaffolds showed significantly higher vascularization and is thus preferable over the use of PCL-collagen I-PEO scaffolds with regard to a future optimization of cell survival in the EPI loop model. The construct developed here shows a promising approach for in vivo implantation of myoblasts and stem cells and takes another step towards the generation of functional skeletal muscle.

## Figures and Tables

**Figure 1 cells-11-03774-f001:**
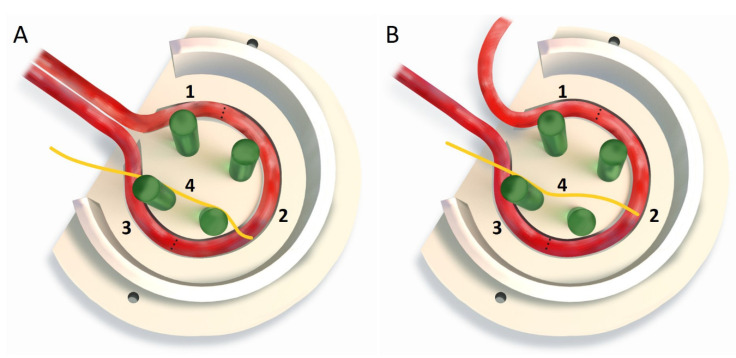
A: Comparison between the newly modified (**A**) and the previously described (**B**) EPI loop model, showing the different entry angles of the arteries into the chamber. EPI loop consisting of SIEA (1), vein interposition (2), SIEV (3) and motoric nerve branch (4) B: EPI loop consisting of saphenous artery (1), vein interposition (2), SIEV (3) and motoric nerve branch (4).

**Figure 2 cells-11-03774-f002:**
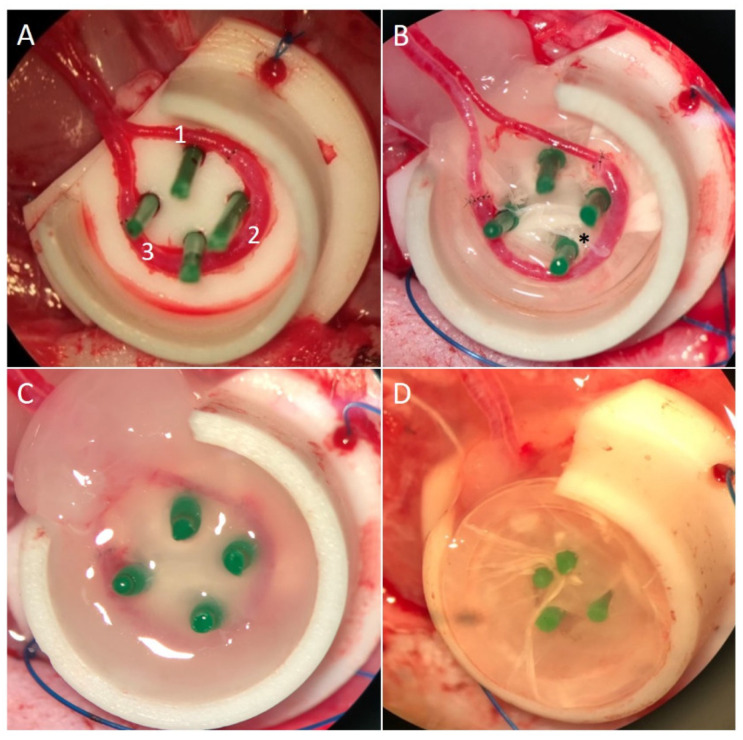
(**A**): EPI loop consisting of SIEA (1), vein interposition (2) and ipsilateral SIEV (3); (**B**): PTFE chamber including four spacers and two layers of nanofiber scaffolds, EPI loop with obturator nerve (*); (**C**): Fibrin gel on top of scaffolds and EPI loop; (**D**): two more layers of PCL-collagen/PCL-collagen-PEO nanofiber scaffolds on top.

**Figure 3 cells-11-03774-f003:**
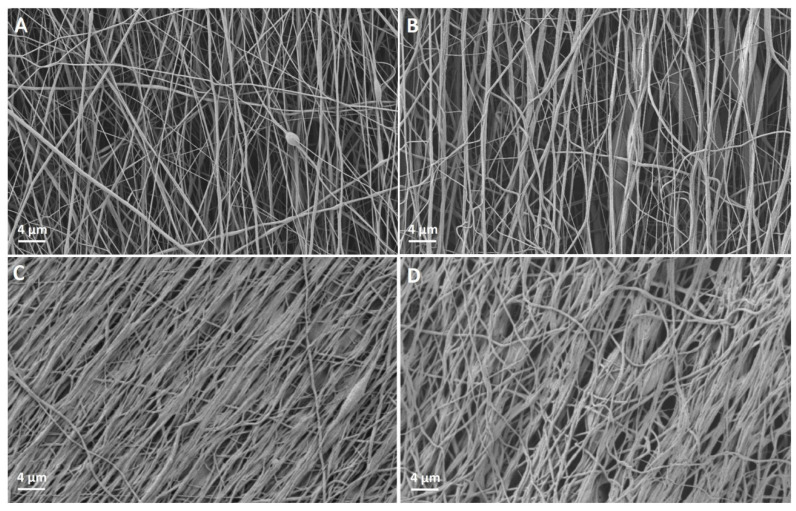
(**A**): PCL-collagen I-nanofibers before pretreatment. (**B**): PCL-collagen I-PEO-nanofibers before pretreatment. (**C**): PCL-collagen I-nanofibers after treatment with EtOH and PBS. (**D**): PCL-collagen I-PEO-nanofibers after treatment with EtOH and PBS.

**Figure 4 cells-11-03774-f004:**
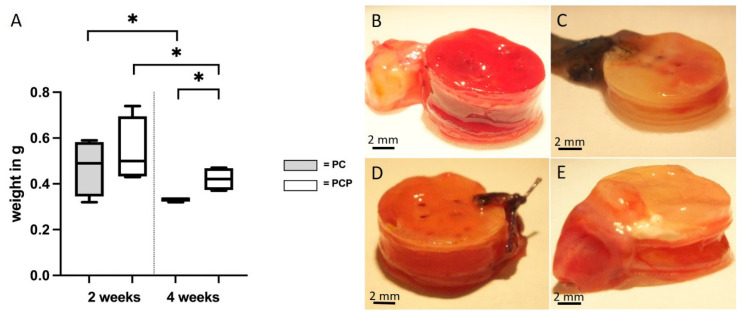
Explanted scaffold-EPI loop-constructs. (**A**): Explant weights. Box plots demonstrate decrease from two weeks to four weeks of implantation and differences of PC and PCP after four weeks of implantation (* *p* ≤ 0.05, unpaired *t*-test); (**B**): PCL-collagen I-scaffold-construct after two weeks of implantation (**C**): PCL-collagen I-scaffold-construct after four weeks of implantation (**D**): PCL-collagen I-PEO-scaffold-construct after two weeks of implantation (**E**): PCL-collagen I-PEO-scaffold-construct after four weeks of implantation.

**Figure 5 cells-11-03774-f005:**
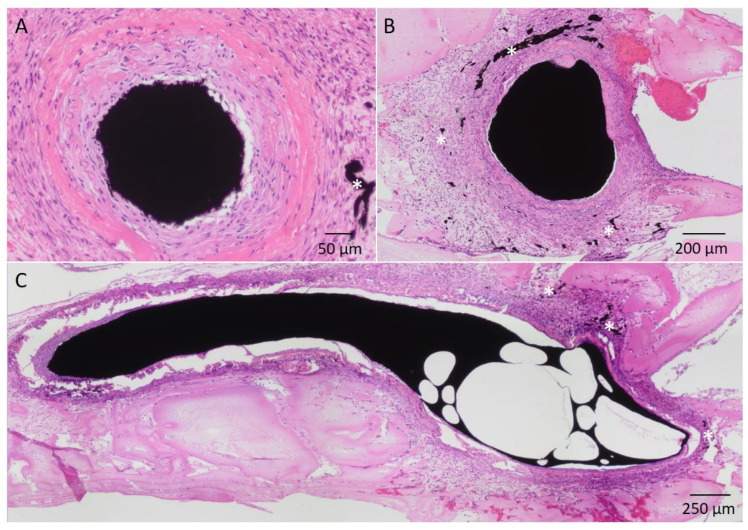
H&E stained sections of a two weeks PCL-collagen I-PEO-construct. Exemplary newly formed vessels are indicated by (*). (**A**): Superficial inferior epigastric artery filled with India ink; (**B**): Superficial inferior epigastric vein filled with India ink, surrounded by newly formed vessels. (**C**): Longitudinal section through the venous interposition filled with India ink, surrounded by newly formed vessels.

**Figure 6 cells-11-03774-f006:**
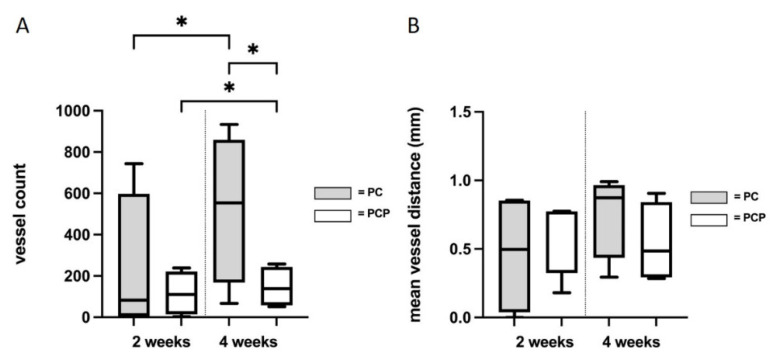
(**A**): Number of newly formed vessels. (**B**): Mean vessel distance of newly formed vessels to the main EPI loop vessels. An unpaired *t*-test or Mann–Whitney test was performed as appropriate. Statistically significant differences are marked with * for *p* ≤ 0.05.

**Figure 7 cells-11-03774-f007:**
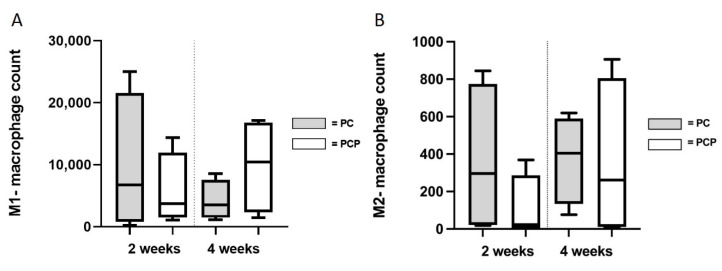
Count of pro- (M1) (**A**) and anti-inflammatory (M2) (**B**) macrophages. No differences in the immunological interaction of PCL-collagen I-scaffolds or PCL-collagen I-PEO-scaffolds were found after two weeks or four weeks.

**Figure 8 cells-11-03774-f008:**
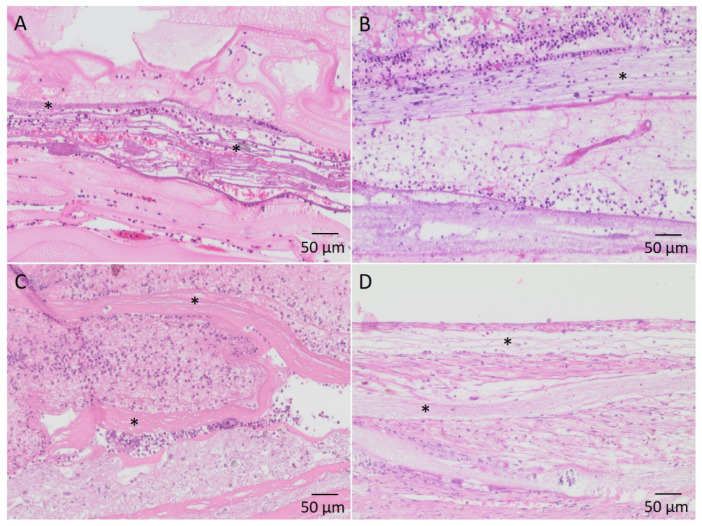
H&E stained sections showing infiltration of cells into the nanofiber scaffold layers. Exemplary nanofibers are marked by (*) (**A**): PCL-collagen I-PEO-scaffolds after two weeks of implantation; (**B**): PCL-collagen I-PEO-scaffolds after four weeks of implantation; (**C**): PCL-collagen I-scaffolds after two weeks of implantation; (**D**): PCL-collagen I-scaffolds after four weeks of implantation.

**Figure 9 cells-11-03774-f009:**
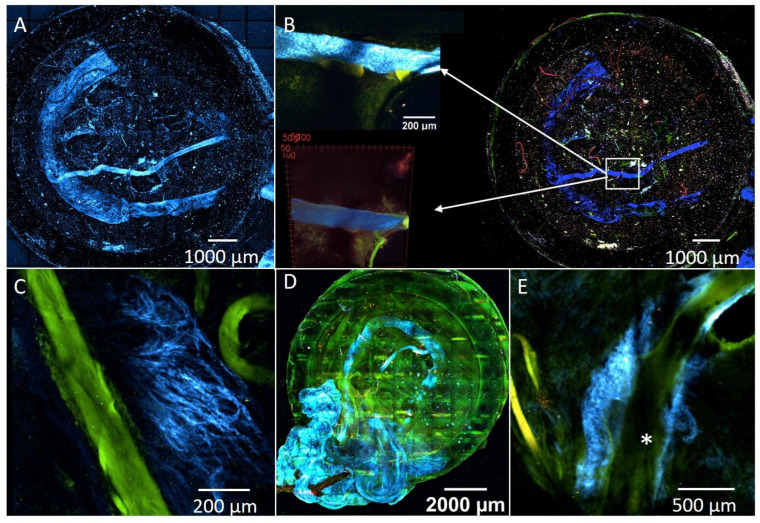
Color pattern: Blue indicates the SHG signal of collagen in vascular and nerve structures. Green and red represent the tissue autofluorescence. (**A**): EPI loop PC construct with subsequent explantation. Strong signal of motor nerve branch and vessels in SHG channel; (**B**): Positive signal of motor nerve branch in SHG channel; (**C**): newly formed capillaries (blue) in PC4; (**D**): Visualization of the EPI loop model (PCP) after four weeks of implantation; (**E**): Neoangiogenesis (blue) in the area of the EPI loop (indicated by (*)).

**Figure 10 cells-11-03774-f010:**
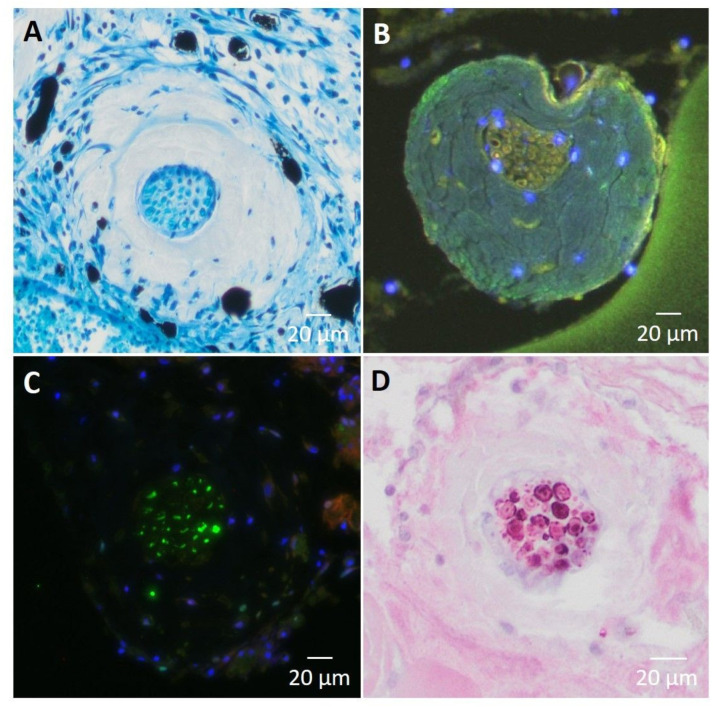
(**A**): Cross section through the motor nerve branch, stained with methylene blue (PCP4). Newly formed vessels filled with India ink show pronounced vascularization in the vicinity of the nerve. (**B**) (PC2) and (**C**) (PCP4): Synaptophysin staining of axon cross-section shows the presence of synaptic vesicle proteins p38 (green). (**D**): S100 immunohistochemistry showing peripheral glial cells (Schwann cells) between the axons of the motor nerve (PCP2).

**Table 1 cells-11-03774-t001:** Number of operated animals (patency rate).

	PC	PCP
2 weeks	7 (57%)	7 (57%)
4 weeks	5 (80%)	8 (50%)

## Data Availability

The data that support the findings of this study are available from the corresponding author upon reasonable request.

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
