# Peer review of "Vascularization of Poly-ε-Caprolactone-Collagen I-Nanofibers with or without Sacrificial Fibers in the Neurotized Arteriovenous Loop Model"

_cells, 2022, doi:10.3390/cells11233774_

Round 1

Reviewer 1 Report

This paper talks about the vascularization and neurotization of axially aligned nanofibrous scaffolds made of polycaprolactone and collagen. Polyethylene oxide used as sacrificial material did not show any efficiency in improving the pore size of the scaffold and keeping the alignment intact. PC scaffolds showed better vascularization and neurotization than PCP scaffolds.

PEO was added as a sacrificial material to improve the pore size of the nanofibrous scaffold. So, scaffold characterizations, including measurement of pore size and porosity, especially after the treatment, should be performed and reported. Also, the fiber diameter and mechanical properties of the scaffolds after treatment should be reported.

2   Ironically, the observed results did not support the hypothesis. The authors brought a few previously published research examples to support the unexpected outcomes. Visually, PC scaffolds had smaller pore sizes but more aligned fibers than PCP scaffolds. In the densely arranged scaffolds, cells face difficulties penetrating the scaffolds. However, PC scaffolds showed better vascularization and neurotization than PCP scaffolds. The explanation provided by the authors in the discussion section is insufficient and does not consider all concerned issues. This should be investigated further or supported by other research work concretely and reported.

3 The authors claim that the EPI loop improved vascularization and neurotization in scaffolds. The vascularization and neurotization in PC and PCP scaffolds without the EPI loop should be included as controls to show the efficiency of the EPI loop, especially when the observed results did not support the hypothesis.

Reviewer 2 Report

Authors present a novel model of combined axial vascularization and neurotization of a PCL-collagen I-nanofiber construct. The advantage of the model over the existing ones is the structure of scaffold, allowing for the improved vascularization in situ, resulting in better survival rate of in vivo implanted myoblasts and stem cells.

The project is concentrated on the improvements in scaffold engineering (e.g. improvement of space for neovascularization inside scaffold by dissolving of PEO and formation of gaps in PCP. Application of sacrificial fibers technique allowed for the improvement of PCL electrospun nanofiber structure, making it more "angiogenesis-friendly". Another novelty is the modification of the EPI loop by replacing the saphenous artery (previously used in the construct) by superficial inferior epigastric artery.

The results are well documented using appropriate research methods, the data confirming the researcher's conclusions are complete and fully convincing.

The novelty and significancy of the manuscript data concern the improvements in scaffold construction for muscle regeneration, which is more material engineering, that cell biology. The decision, if such topic fits the profile of the Special Issue "Emerging Topics in Vascular Endothelial Cell Biology" is the privilege of the Special Issue Editor.

Round 2

Reviewer 1 Report

Included in the attached file.
